# Prevalence of High-Risk Disordered Eating Amongst Adolescents and Young Adults in the Middle East: A Scoping Review

**DOI:** 10.3390/ijerph19095234

**Published:** 2022-04-26

**Authors:** Mahmoud Azzeh, Gemma Peachey, Tom Loney

**Affiliations:** 1College of Medicine, Mohammed Bin Rashid University of Medicine and Health Sciences, Dubai P.O. Box 505055, United Arab Emirates; mahmoud.azzeh@students.mbru.ac.ae (M.A.); gemma.peachey@slam.nhs.uk (G.P.); 2South London and Maudsley NHS Foundation Trust, The Maudsley Hospital, Denmark Hill, London SE5 8AZ, UK

**Keywords:** anorexia nervosa, binge-eating disorder, bulimia nervosa, epidemiology, feeding and eating disorders, Middle East, prevalence

## Abstract

High-risk disordered eating (HRDE) negatively affects physical, mental, and social wellbeing. This scoping review aimed to estimate the prevalence of HRDE amongst adolescents and young adults in the Middle East. MEDLINE database was searched for studies published in English or Arabic from 1 January 2000 to 30 September 2020, estimating HRDE prevalence (using the Eating Attitudes Test 26 or 40 item questionnaire) in the Middle East. Two reviewers independently screened abstracts and full texts of potentially eligible records, followed by data extraction from eligible studies. Nineteen studies (*n* = 16,288; 65.8% female) from Egypt, Iran, Israel, Jordan, Kuwait, Libya, Oman, Palestine, Saudi Arabia, Syria, Turkey, and the United Arab Emirates were included. Prevalence of HRDE varied considerably across countries and was lowest amongst adolescents in Israel (F 8.2%; M 2.8%) and highest amongst university students in Egypt (F 75.8%; M 69.6%). Prevalence of high-risk for anorexia nervosa ranged from 0.0% in Jordan to 9.5% in Oman; high-risk for bulimia nervosa from 0.6% in Jordan to 1.0% in the United Arab Emirates; and high-risk for binge eating disorder was 1.0% and 1.8% in Turkey and Jordan, respectively. Future studies should employ a standardized two-stage design with clinical diagnosis to verify the prevalence of abnormal eating behaviours in the Middle East.

## 1. Introduction

Eating disorders (ED) are defined as “a disturbance in eating habits that result from either excessive or insufficient food intake” [1]. Globally, the three most common eating disorders are anorexia nervosa, bulimia nervosa, and binge eating disorder [2]. Anorexia nervosa (AN) is defined as a disturbance in the way one’s body weight or shape is experienced, undue influence of body shape and weight on self-evaluation, or persistent lack of recognition of the seriousness of the current low body weight. Bulimia nervosa (BN) is characterized by recurrent inappropriate compensatory behaviour in order to prevent weight gain, such as self-induced vomiting, misuse of laxatives, diuretics, or other medications, fasting, or excessive exercise. Binge eating disorder (BED) is characterised as eating in a discrete period of time (e.g., within any 2 h period) an amount of food that is definitely larger than what most people would eat during a similar period of time and under similar circumstances [2]. Anorexia nervosa has one of the highest mortality rates (5.9 per 1000 person-years) for any psychiatric disorder, whereas BN has a lower mortality rate of 1.7 per 1000 person-years [3]. The exact aetiology of ED is still unknown with research suggesting the interplay of a combination of factors, including family relationships, psychological issues, and genetics [1]. However, using multimedia technologies and poor body image perception are considered two of many important predisposing factors [4], whereas culture and religious beliefs have also been hypothesized to affect the prevalence of eating disorders in a given community [5].

A two-stage study design is the gold standard method for estimating the prevalence of eating disorders in a population [6]. The first stage involves the study sample completing a psychometrically validated self-administered questionnaire, such as the Eating Attitude Test-40 (EAT-40) or its shorter version (EAT-26) [7]. Questionnaire scores identify participants that might be at a high risk of developing or having an eating disorder. During the second stage, participants classified as high risk complete a clinical assessment interview with a psychiatrist to ascertain the diagnosis of a specific eating disorder. The term “high risk of ED” is also known as “disordered eating attitude” or “high risk of disordered eating”; in this paper with have decided to use the term “high-risk disordered eating” (HRDE).

In 2010–2011, the ARAB-EAT project assessed obesity, eating attitudes, barriers to healthy eating, and physical activity amongst adolescents (*n* = 4698; aged 15–18 years) in seven cities in Arab countries, namely Algeria, Jordan, Kuwait, Libya, Palestine, Syria, and the United Arab Emirates [8]. This study reported that disordered eating among adolescents in Arab countries was relatively high amongst male and female adolescents (ranging from 13.8% to 47.3% among males; 16.2% to 42.7% among females) [8]. Adolescence and early adulthood are considered important risk periods for ED development. Moreover, the study suggested that the prevalence of disordered eating amongst Arab adolescents was similar to prevalence estimates for adolescents living in western developed countries, but twice as high compared to the same age group in Asian countries [7]. Early detection and intervention of eating disorders or high-risk disordered eating is imperative for early intervention since a longer duration of eating disorders is associated with lower chances of successful treatment [9]. Additionally, when diagnosing AN and BN at or before the age of 19 years, adolescents are four to eight times more likely to recover from AN and BN, respectively [10]. Despite the public health importance of eating disorders, there has not been a systematic evaluation and synthesis of the literature on the prevalence of eating disorders in the Middle East. The objective of this scoping review was to estimate the prevalence of high-risk disordered eating (HRDE) amongst adolescents and young adults in the Middle East.

## 2. Materials and Methods

This paper has been written in accordance with the Preferred Reporting Items for Systematic reviews and Meta-Analyses (PRISMA) guidelines [11].

### 2.1. Study Design

A scoping review was conducted on studies reporting the prevalence of disordered eating amongst adolescents and young adults in the Middle East. The study was conducted in accordance with the Declaration of Helsinki and approved by the Institutional Review Board of the Mohammed Bin Rashid University of Medicine and Health Sciences (MBRU-IRB-SRP2018-015 08 March 2018).

### 2.2. Eligibility Criteria

Studies were included if they: (1) were published in English or Arabic (or had an available English translation) between 1 January 2000 and 30 September 2020; (2) used the 26-item or 40-item Eating Attitudes Test (EAT-26; EAT-40) with or without clinical assessment, to assess the prevalence of high risk disordered eating [7]; and (3) recruited a sample from a country in the Middle East (defined as Bahrain, Cyprus, Egypt, Iran, Iraq, Israel, Jordan, Kuwait, Lebanon, Libya, Oman, Palestine, Qatar, Saudi Arabia, Syria, Turkey, United Arab Emirates, and Yemen). No exclusions for age, sex, or setting were applied. Both cross-sectional and longitudinal studies were eligible for inclusion.

### 2.3. Exclusion Criteria

Studies that did not meet all three criteria were excluded. Books, editorials, reviews and qualitative studies, i.e., studies measuring the prevalence of disordered eating with questionnaires other than EAT-26 or EAT-40 (e.g., Eating Disorders Inventory; EDI), Bulimic Investigatory Test, Edinburgh; BITE) were excluded to maximise homogeneity. Studies assessing Pica, rumination disorder, and any unspecified feeding or eating disorder were omitted.

### 2.4. Information Sources and Literature Search

The literature search was performed in MEDLINE and the reference lists of each retrieved paper were searched manually for additional studies. Other websites such as the World Health Organization–Regional Office for the Eastern Mediterranean (WHO-EMRO) website and mental health websites were also searched manually for possible eligible papers. A manual search by the authors’ name of all retrieved studies to identify additional studies was also conducted. The literature search was last updated on the 30 September 2020. In collaboration with a research librarian, we conducted a broad literature search using a combination of MeSH search terms and keywords to minimize the likelihood of missing evidence:

(“Feeding and Eating Disorders”[Mesh] AND (“Middle East”[Mesh] OR “Cyprus”[Mesh] OR “Egypt”[Mesh]) AND “Prevalence”[Mesh] AND (“2000/01/01”[PDAT]:“2020/09/30”[PDAT])).

We also searched Internet search engines and hand-searched the reference lists of previous systematic reviews on the topic and of all included studies.

### 2.5. Study Selection

Two independent reviewers (MA and TL) manually screened the titles and abstracts of studies retrieved from the search and duplicates were removed. Studies considered eligible for full text screening were retrieved for full text review. Any conflicts were resolved by a third independent reviewer (GP), who is also a psychiatrist.

### 2.6. Data Collection Process and Data Items

Data from each paper satisfying the inclusion criteria was extracted manually by one reviewer (MA) into a pre-defined Excel file. One reviewer (TL) independently double-checked the accuracy of the data extracted that was entered into a summary table in the following categories: Authors, Year of Publication, Country, Setting, Sample Size (Response Rate), Sex Distribution (%), Age Group or Mean Age (Years), Questionnaire Used (Language), Prevalence (%) of High-Risk Disordered Eating, Clinical Assessment (Yes/No and Type), and Eating Disorders Prevalence (%).

#### The Eating Attitudes Test

The Eating Attitudes Test is available as a long-form 40-item (EAT-40) or short-form 26-item (EAT-26) [7] self-administered questionnaire used for screening large populations for attitudes and symptoms characteristic of eating disorders in hospitalized patients as well as the general population [12]. The EAT questionnaires are comprised of three sections; (1) an individual’s BMI; (2) 26- or 40-EAT questions on a six-point Likert scale ranging from *‘never’* to *‘always’* (e.g., *‘I am terrified about being overweight’*); and (3) behavioural questions regarding binge eating and purging habits (e.g., *‘Lost 20 pounds or more in the past 6 months’* ‘Yes’ or ‘No’). The cut-off scores for the EAT-26 and EAT-40 questionnaires are 20 and 30, respectively, which indicates the individual completing the questionnaire has a high-risk of having an eating disorder. BMI and the behavioural questions are risk factors for ED, both requiring clinical assessment if answered positively. The EAT-26 and EAT-40 are the most commonly used tools to assess the prevalence of high risk disordered eating in clinical and non-clinical settings due to their high reliability and validity [7]. The EAT-26 is formed of three sub-scales: dieting (13 items), bulimia and food preoccupation (six items), and oral control (seven items). A high score alone does not yield a specific diagnosis of an eating disorder. Scores ≥ 20 for EAT-26 and ≥30 for EAT-40 indicate a need for further investigation of body mass history, current body mass index, and percentage of ideal body weight, by a qualified professional.

### 2.7. Summary Measures 

This scoping review conducted a narrative synthesis of the included studies and reported the estimated prevalence of high-risk disordered eating by study, country, and overall.

## 3. Results

### 3.1. Study Selection and Characteristics

The PRISMA flowchart in Figure 1 summarises the scoping review process. The initial MEDLINE search strategy yielded 50 titles and two additional papers were identified from other sources (Figure 1). Ineligible and duplicate studies were removed and 19 papers from 13 countries that satisfied the inclusion criteria were included in the final review (Figure 1). No studies meeting the inclusion criteria were published between 2000 and 2002. The key information from the included studies (2002–2020) is summarized in Table 1. Studies are grouped by countries of the Middle East and listed in chronological order.

### 3.2. Study Characteristics

Included studies consisted of six (32%) papers from Turkey; three (16%) from Israel; two each from Iran (11%), Egypt (11%), and the United Arab Emirates (UAE; 11%); one each from Jordan (5%), Oman (5%), and Syria (5%); and one study (5%) reported data from Jordan, Kuwait, Libya, Palestine, Syria, and the UAE. No studies were found for Bahrain or Yemen. In addition, no studies from Cyprus, Lebanon, or Qatar met the eligibility criteria. A total of six studies reported prevalence estimates in females only whereas one study reported prevalence estimates in males only. The 12 remaining studies reported prevalence estimates in both males and females. Participant age ranged from 10–51 years and there was no trend in the age selection in the studies. Only one study specifically included a sample of adults above 30 years. The majority of papers mentioned the age group and/or the mean age, except one paper, which did not report the age characteristics of the sample.

All the included papers were cross-sectional studies and the sample sizes ranged from 100 to 2907 with the majority of studies (*n* = 14; 74%) reporting HRDE prevalence estimates from samples ranging in size from 100 to 800. Four studies presented data from samples ranging from 800 to 2907. One study included samples from seven different countries and the sample size for each country ranged from 477 to 1062. The response rate varied between 64% and 100%; however, eight studies did not include participant response rates. The majority of studies were conducted in educational settings with twelve studies in middle and high schools and one study that involved primary school students. Six studies were conducted in universities whereas one study was completed within a fitness program and one within a weight-loss centre.

Only three (16%) of the included studies utilized the two-stage ‘gold standard’ design used to estimate the prevalence of disordered eating, namely, the use of a psychometrically validated self-administered questionnaire to identify participants at high risk of disordered eating, followed by a clinical assessment. In addition, one study provided the prevalence of eating disorders (AN, BN, and BED) using a questionnaire that incorporated criteria from DSM-IV-TR to diagnose ED but with no clinical assessment. The remainder of the studies employed EAT-26 or EAT-40 to assess high risk of eating disorders in their samples. The abbreviated EAT-26 questionnaire was used in 52% of the included studies to assess high risk of disordered eating and translated versions included Arabic, English, Hebrew, and Persian. Seven studies used EAT-40 in Arabic, Hebrew, and Turkish.

## 4. Discussion

### 4.1. Summary of Major Findings

The current study is the first scoping review of eating disorders and the high risk of eating disorders in the Middle East between 2000 and 2020. The review found consistency across studies in terms of study design (cross-sectional using a self-administered questionnaire with no clinical assessment) and study setting (e.g., educational), but varied widely with regard to sample characteristics (e.g., age and sex distribution), outcome ascertainment (e.g., language and version of EAT used), and reported prevalence. Turkey was the country with most studies (*n* = 6); 66% of the studies in Turkey were conducted in universities using the Turkish EAT-40. Three studies were conducted in both the UAE and Israel in high school settings. Egypt, Iran, and Jordan conducted two studies each and a third of these studies were conducted in female populations only. Kuwait, Libya, Oman, Palestine, Saudi Arabia, and Syria had one study each. None of the included studies were from Bahrain, Cyprus, Iraq, Lebanon, Qatar, or Yemen. Four studies reported prevalence estimates for eating disorders in the Middle East and anorexia nervosa was the most common ED studied (*n* = 4). The prevalence of AN was 0%, 0.03%, 1.0% and 9.5% in Jordan, Turkey, the UAE, and Oman respectively [20,21,27,29]. Bulimia nervosa was reported to have a prevalence of 0.6%, 0.8%, and 1.0% in studies conducted in Jordan, Turkey and the UAE, respectively [20,27,29]. Lastly, binge eating disorder was reported in Turkey and Jordan, with a prevalence of 1.0% and 1.8%, respectively. However, it was unclear whether the study in Jordan used a clinical interview in their study design, as they utilized the Eating Habits Questionnaire (EHQ) to detect ED, whereas identification of AN, BN, and BED was based on DSM-IV-TR criteria. One study conducted in the U.S. on a nationally representative sample revealed the lifetime prevalence of AN, BN, and BED to be 0.80%, 0.28%, and 0.85%, respectively [31]. In addition, AN prevalence ranged between 1.2% and 4.2% [32] and the point prevalence for BN is generally ~1% [33], with the lifetime prevalence between 1.7% and 2.9% in Australia, Finland, and Sweden [32]. Previous work has reported that the prevalence of binge eating disorder ranged from 0.3% to 3.5% in European countries and in the U.S. [33,34,35,36].

The EAT-40 is currently the most commonly used questionnaire to assess high risk of ED and disordered eating attitudes worldwide [19]. High risk of ED was assessed in all the included studies using different versions of EAT-26 and EAT-40 in Arabic, English, Hebrew, Persian, and Turkish, most of which were validated through pilot studies within cross-sectional studies or through other papers [15,37,38,39]. High risk of ED ranged from 6.1% to 73.3% in our review, which might reflect ‘true’ differences in the prevalence of high-risk disordered eating, or could possibly be due to differences in interpretation of the questionnaire items. A previous study recruited a representative sample of female students in grades 7–12 (*n* = 129) in Riyadh, Saudi Arabia (1998–1999), and reported that nearly a fifth (19.3%) were identified by the EAT-26 as having abnormal eating attitudes, with only one case of AN and no cases of BN [37]. The only study from Saudi Arabia in our review reported that a third (35.4%) of female University students (aged 17–33 years) were classified with high-risk disordered eating using the EAT-26 (Arabic). The prevalence of high-risk disordered eating tends to peak during late adolescence and early adulthood [40]. Therefore, the difference in prevalence estimates between these two studies could possibly be related to the different age groups sampled.

### 4.2. Comparison with Prevalence Estimates from Other Countries

Prevalence estimates for ED in countries outside of the Middle East region are generally lower than reported in our review. A study in Algeria reported 15.2% [8], while in Japan, South Korea, and Taiwan, the prevalence of high-risk ED was 10.3%, 11.2%, and 10.4%, respectively [41,42,43]. Studies conducted in Canada, Spain, the UK, and the US reported that the prevalence of high-risk ED ranged from 11% to 26% [44].

There is no clear explanation of the factors leading to a higher prevalence of high risk of ED in the Middle East region compared to other countries, and it is currently unclear whether this is due to a higher prevalence of ED risk factors or methodological issues. Numerous studies in our review reported methodological limitations, including the absence of the gold standard two-stage design for assessing ED and the recruitment of small and/or non-representative samples due to limited funding and resources available for this type of research [18,20]. Moreover, it is challenging to compare prevalence estimates between and within countries as the time period for data collection and population characteristics differ widely in terms of composition (age, sex, and ethnicity). Previous work has suggested that the prevalence of ED is different in urban versus rural areas with a higher BN prevalence in the urban population [45,46]. Numerous studies have explored the association between various factors (e.g., diabetes Type 1, sexual abuse, BMI, family status, parental educational level, family history of ED, and socioeconomic status) and ED [20,23,24,27,29]; however, there are no definite factors accounting for the higher prevalence of disordered eating in the Middle East region. The aetiology is not fully understood but it is hypothesized that it is an interaction between genetic, cultural, and psychological factors that drives individuals towards eating disorders [47].

Previous work has reported that BN, but not AN, is a culturally bound disorder and both disorders are more prevalent in industrialised and westernised societies [47]. In view of global economic developments and sociocultural changes, it is not surprising that the prevalence of such disorders has increased in the Middle East region [47,48]. Traditionally, the general perception of the preferred body type in the Arab culture was for women to be plump [49]. However, recent studies have revealed that Middle Eastern adolescent females are more likely to strive for thinness [50,51]. In parallel, obesity levels are on the rise globally and the Middle East is no exception with countries from the region occupying 11 out of the top 30 countries with the highest prevalence of obesity in the world [52]. Currently, there is a lack of data on the perceptions of body ideals or beauty standards of male adolescents and young adults in the Middle East, and future mixed-methods studies might be required to explore the sociocultural drivers of ED development in this group.

### 4.3. Strengths, Limitations, and Future Directions

This is the first study to review the prevalence of high-risk disordered eating in the Middle East. The review included articles in both Arabic and English over a 20-year period; however, as with any review, it is possible that studies in other languages not retrieved by our search strategy were missed. Cautious interpretation is required when comparing HRDE prevalence estimates between studies due to clinical (varying age and sex composition of samples) and methodological (different EAT versions and one- versus two-step design) heterogeneity. Moreover, the EAT-26 and EAT-40 are self-report diagnostic screening tools that indicate a high risk for disordered eating rather than a specific diagnosis of an eating disorder. High scores on these self-report tools require further investigation by a qualified professional, in combination with body mass history, current body mass index, and percentage of ideal body weight.

The majority of studies (87.5%; *n* = 17) were conducted in educational settings, such as high schools and universities; however, some studies have included heterogenous population in terms of age (adolescents, young adults, and middle-aged adults). Additionally, there were two studies that recruited participants from a weight loss center in Egypt [14] and from a fitness programme in Iran [18]. The contextual setting of the study may have created a potential selection bias, leading to higher self-reported HRDE. The included studies used versions of EAT-26 and EAT-40 in different languages, and the prevalence estimates for high-risk disordered eating could be partially related to differences in interpretation of the questionnaire items, which may be prone to bias due to their self-reported format. The EAT-26 and EAT-40 might have limitations when applied to populations of different cultures [53]. In addition, the EAT-26, EAT-40 and AN criteria in DSM IV has a component pertaining to loss of period which can underestimate the prevalence of males suffering from HRDE and ED [25]. Finally, the EAT-26 and EAT-40 are not sensitive to binge eating disorder.

Despite it not being a previously common practice to register and publish the protocols for scoping reviews such as this one, we support this initiative for future scoping reviews as it provides full transparency on the scoping review process. Due to the low volume of articles retrieved from the literature search, it was not deemed necessary to use reference or systematic review management software (e.g., Endnote, Covidence) for the screening or management of the scoping review. However, the use of such software in future scoping or systematic reviews with a larger volume of retrieved articles might make this stage of the review process more efficient.

Future research should adopt the two-stage design to report the prevalence of ED. Furthermore, studies should strive to adopt the DSM-5 criteria for ED, which excludes amenorrhea from its criteria for diagnosing AN and improves the suitability of the criteria for males. Moreover, standardising methodology and outcome ascertainment will improve consistency in the results, which might allow researchers to compare prevalence estimates within and between countries. If the issue of low funding for ED research persists, researchers could benefit from technological innovations in the field of diagnosing ED. For example, Eating Disorder Assessment for DSM 5 (EDA-5), which is an app-based diagnostic instrument, proved to be equally as effective as clinical interviews in diagnosing AN, BN, and BED [54]. Moreover, longitudinal studies starting during the pre-pubescent period with regular follow-up during adolescence and early adulthood would provide vital information on the incidence and factors associated with ED, in addition to the critical time periods of life when ED develops and manifests. Such information may be useful to public health professionals and organisations to develop awareness, screening, and intervention programmes in schools, universities, and in the general community. Overall, eating disorders are an under-recognized public health issue in the Middle East region and patients typically prefer to be supported by family and friends rather than seek medical services which are not always readily available.

## 5. Conclusions

This is the first scoping review of disordered eating in the Middle East region that found a relatively large proportion of adolescents and adults were classified as high risk of ED. Nonetheless, caution is required when comparing prevalence estimates between studies and countries. The current body of disordered eating research in the Middle East is scant and harmonised methodology on larger representative samples using the two-step design will help to provide more accurate, valid, and reliable ED prevalence estimates in the region.

## Figures and Tables

**Figure 1 ijerph-19-05234-f001:**
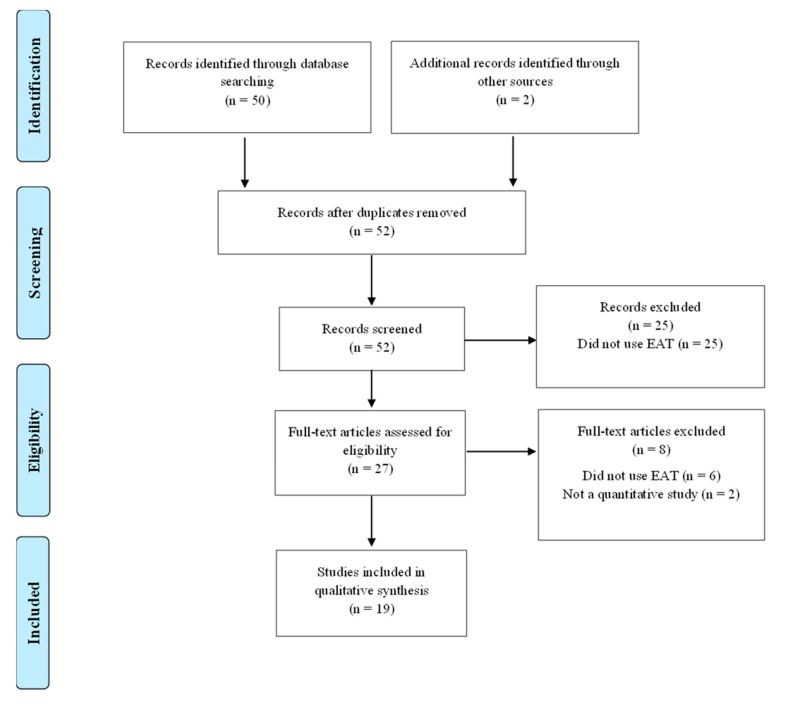
PRISMA Flow diagram of the search and selection process.

**Table 1 ijerph-19-05234-t001:** Overview of studies assessing the prevalence of high-risk disordered eating in Middle Eastern countries from 2000–2020. Studies are grouped by country and listed in chronological order.

Author (Year)	Country	Setting	Sample Size (Response Rate)	Sex (%)	Age Group/Mean Age (Years)	Test Used (Language)	Prevalence of High-Risk Disordered Eating (%)	Clinical Assessment Yes/No	Eating Disorders Prevalence (%)
El-Bagoury et al. (2018) [13]	Egypt	University	445 (NA)	59.3% 	(17–26) 20.3±1.50	EAT-26 (Arabic)	73.3% T 75.8%  69.6% 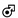	No	NA
Eladawi, et al. (2018) [14]	Egypt	Weight Loss Centre	400 (88.0%)	72.0% 	(NR) 35.2 ± 11.6	EAT-40 (NA)	65.0% T 66.7%  61.0% 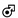	No	NA
Gargari, et al. (2010) [15]	Iran	Fitness Program	250 (92.6%)	100% 	(14–51) NR	EAT-26 (Persian)	28.4%	No	NA
Jalali-Farahani et al. (2015) [16]	Iran	High School	465 (95.8%)	48.8% 	(NR) 15.6 ± 0.9	EAT-26 (English)	18.9% T 26.4%  11.8% 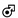	No	NA
(Maor et al. (2006) [17]	Israel	High School	283 (78.0%)	51.0% 	(NR) NR	EAT-26 (Hebrew)	13.1% T 20.8%  5.0% 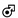	No	NA
Latzer et al. (2009) [18]	Israel	Middle, High School	1141 (NA)	100% 	(12–18) NR	EAT-26 (Arabic) EAT-26 (English)EAT-26 (Hebrew)	25.0%	No	NA
Katz (2014) [19]	Israel	Middle, High School	323 (99.1%)	56.0% 	(NR) 14.4 ± 1.3	EAT-40 (Hebrew)	6.1% T 8.2%  2.8% 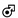	No	NA
Mousa et al. (2010) [20]	Jordan	Primary, Middle, High Schools	326 (75.5%)	100% 	(10–16) 12.9 ± 1.8	EAT-26 (Arabic)	40.5%	Yes (EHQ)	0% AN 0.6% BN 1.8% BED
Al-Adawi et al. (2002) [21]	Oman	Middle, High School	262 (NA)	48.0% 	(NR) 15.4±1.4	EAT-26 (Arabic)	29.0% T 53.0%  47.0% 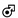	Yes (CIDI)	9.5% AN
Abd El-Azeem Taha et al. (2018) [22]	Saudi Arabia	University	1200 (NA)	100% 	(17–33) NR	EAT-26 (Arabic)	35.4%	No	NA
Elal et al. (2004) [23]	Turkey	University	532 (NA)	100% 	(NR) 19.9±1.7	EAT-40 (Turkish)	9.8%	No	NA
Pinar (2005) [24]	Turkey	High School	100 (64.3%)	50.0% 	(12–18) 15.5± 1.4	EAT-40 (Turkish)	43.0%	No	NA
(Şanlier et al. (2008) [25]	Turkey	University	610 (NA)	44.6% 	(17–23) NR	EAT-40 (Turkish)	22.8% T 25.7%  20.4% 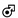	No	NA
Tozun et al. (2010) [26]	Turkey	University	679 (91.1%)	27.5% 	(17–29) 21.6 ±2.2	EAT-40 (Turkish)	6.8% T 9.1%  5.9% 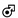	No	NA
Vardar & Erzengin (2011) [27]	Turkey	High School	2907 (100%)	54.0% 	(NR) 17.04 ± 0.8	EAT-40 (Turkish)	8.0%	Yes (SCID)	0.034% AN 0.79% BN 1.0% BED
Sanlier et al. 2016) [28]	Turkey	University	900 (85.7%)	58.0% 	(17–23) 20.4 ± 1.7	EAT-40 (Turkish)	22.8% T 25.7%  20.4% 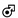	No	NA
Eapen et al. (2006) [29]	UAE	Middle, High School	495 (99.0%)	100% 	(13–18) NR	EAT-40 (Arabic)	23.4%	Yes (KSADS)	1.0% AN 1.0% BN
Musaiger et al. (2013) [8]	Jordan Kuwait Libya Palestine Syria UAE	High School	937 (NA) 628 (NA) 630 (NA) 477 (NA) 1062(NA) 505 (NA)	52.3% 	(15–18) NR	EAT-26 (Arabic)	31.6% T, 32.7%  , 20.1% 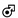 44.7% T, 42.8%  , 47.3% 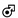 26.7% T, 32.6%  , 19.3% 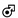 31.7% T, 38.9%  , 23.2% 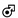 22.9% T, 32.0%  , 14.6% 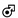 33.5% T, 37.4%  , 29.8% 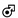	No	NA
Musaiger et al. 2014) [30]	UAE	High School	731 (NA)	100% 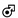	(15–18) NR	EAT-26 (Arabic)	41.2%	No	NA

Notes: 

 = Females; 
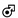
 = Males. AN = Anorexia Nervosa; BN = Bulimia Nervosa, BED = Binge Eating Disorder; EHQ = Eating habits questionnaire; CIDI = Composite International Diagnostic Interview; EAT = Eating Attitude Test; ED = Eating Disorders; KSADS = Schedule for Affective Disorders and Schizophrenia (KSADS) DSM-IV; NA = Not Applicable; NR = Not Reported; SCID = Structured Clinical Interview for DSM-III-R; T = Total.

## Data Availability

Data sharing is not applicable to this article.

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
