# Peer review of "Prevalence of High-Risk Disordered Eating Amongst Adolescents and Young Adults in the Middle East: A Scoping Review"

_ijerph, 2022, doi:10.3390/ijerph19095234_

Round 1
Reviewer 1 Report
Thank you for the opportunity to review your very important work. This study was a review of literature investigating prevalence of eating disorders in Middle Eastern Countries. This work adds to the existing literature on prevalence in this region of the world, and, per author statement, is the first scoping review of this topic in the Middle East. Utilizing the PRIMSA guidelines enforced solid methodological approach to the aims of this study. I only have a few minor points to consider:
- Opioid use disorder actually has the highest mortality rate of any psychiatric disorder, updated recently (higher than AN).
- All but one study in the review was in a school setting (University, high school, etc) and reported outcomes in adults over the age of 30, this should be noted in the limitations
- The biggest challenge is that the EAT-26 is non-diagnostic but rather is a screening tool. You use the term "high risk disordered eating" which is an accurate description of the measure but then in the discussion and summary of findings you report prevalence rates. I would temper the language to state that they were high risk for AN, but it's not appropriate to report prevalence of AN, BED, etc. using solely the EAT-26
Author Response
Reviewer 1
Thank you for the opportunity to review your very important work. This study was a review of literature investigating prevalence of eating disorders in Middle Eastern Countries. This work adds to the existing literature on prevalence in this region of the world, and, per author statement, is the first scoping review of this topic in the Middle East. Utilizing the PRIMSA guidelines enforced solid methodological approach to the aims of this study.
Author Response: We thank Reviewer 1 for taking the time to review our manuscript and providing supportive feedback.
I only have a few minor points to consider:
Opioid use disorder actually has the highest mortality rate of any psychiatric disorder, updated recently (higher than AN).
Author Response: Thank you for your insightful comment. We have amended the text on Page 1 Lines 41-42 accordingly.
All but one study in the review was in a school setting (University, high school, etc) and reported outcomes in adults over the age of 30, this should be noted in the limitations.
Author Response: Thank you for highlighting this observation. We have included two sentences in the ‘Limitations’ section briefly discussing the two studies that recruited participants from a weight loss centre and fitness programme (Page 11 Lines 298-303).
The biggest challenge is that the EAT-26 is non-diagnostic but rather is a screening tool. You use the term "high risk disordered eating" which is an accurate description of the measure but then in the discussion and summary of findings you report prevalence rates. I would temper the language to state that they were high risk for AN, but it's not appropriate to report prevalence of AN, BED, etc. using solely the EAT-26.
Author Response: We have standardised the use of the term high-risk disordered eating and added text about the limitations of the EAT tool in diagnosing an eating disorder (Page 4 Lines 145-148 and Page 11 Lines 293-297).
Reviewer 2 Report
The present study aims to review the literature on the prevalence of clinical eating pathology in countries in the Middle East. While the topic is of importance, the manuscript falls short of being a useful addition to the field.
Major issues:
- The largest issue is there is very little interpretation of the study findings. The entire first paragraph of the discussion simply restates the results. There are no hypotheses presented on what to make of the similarities and differences across or within countries. For example, Pinar (2005) found a prevalence of 43% in high school students in Turkey, whereas Vardar & Erzengin (2011) found a prevalence of 8% also in high school students in Turkey. It is unclear what the reader is intended to take away from the data presented.
- It is also unclear why this manuscript is not a meta-analysis. More interesting findings could be reported by incorporating these analyses. Authors could explore potential moderators including age and sex, rather than stating that these differences obscure one’s ability to draw conclusions.
- The title is misleading. The manuscript does not explore the burden or impact of eating disorders, but rather the prevalence of clinically significant disordered eating. The title should be re-worded to align more specifically with the aims of the manuscript.
Minor issues:
- Two of the studies reported the prevalence of high-risk DE in a weight loss center and a fitness program, do the authors not believe these estimates may be biased by the recruited population?
- Table 1 is not formatted consistently. For example, the “Age group/mean age” column is arranged differently across studies. There is no definition provided for “T” under the “Prevalence of High-Risk Disordered Eating” section. It appears the authors meant to include the type of assessment in the “Clinical Assessment” column but that information was not provided. Further, the type of clinical assessment used seems necessary to specify.
- Line 59 and 60 mention a higher prevalence is suggested in female adolescents, but the evidence provided indicates a similar range for males and females.
- Research demonstrating that the EAT-26 and EAT-40 have been validated for use in Arabic, Turkish, Hebrew, and Persian languages need to come sooner in the manuscript and be described in more detail. Differences in prevalence may be due to differences in interpretation of the questionnaire items.
- It is unclear why “records excluded n= 25 papers” in Figure 1 were excluded. Could more detail be added to the figure? Further, 6 papers were excluded due to not using the EAT-26 or EAT-40; however, those papers could be included if the scales used also have a validated clinical cut-off score.
Author Response
Reviewer 2
The present study aims to review the literature on the prevalence of clinical eating pathology in countries in the Middle East. While the topic is of importance, the manuscript falls short of being a useful addition to the field.
Author Response: Thank you for taking the time to review our manuscript. The insightful feedback has helped us to improve our manuscript and we hope that you think the revised version makes a useful addition to the sparse literature on high-risk disordered eating in the Middle East region.
Major issues:
The largest issue is there is very little interpretation of the study findings. The entire first paragraph of the discussion simply restates the results. There are no hypotheses presented on what to make of the similarities and differences across or within countries. For example, Pinar (2005) found a prevalence of 43% in high school students in Turkey, whereas Vardar & Erzengin (2011) found a prevalence of 8% also in high school students in Turkey. It is unclear what the reader is intended to take away from the data presented. It is also unclear why this manuscript is not a meta-analysis. More interesting findings could be reported by incorporating these analyses. Authors could explore potential moderators including age and sex, rather than stating that these differences obscure one’s ability to draw conclusions.
Author Response: Thank you for your comment. The primary purpose of a scoping review is to synthesize the available evidence on a specific topic, highlight the gaps, and identify the limitations. A scoping review is preferred when the extant scientific literature on a particular topic has not been comprehensively reviewed and/or is heterogeneous in nature and not suitable for a more precise systematic review. We think that our scoping review synthesizes interesting estimates on the prevalence of high-risk disordered eating in the Middle East whilst acknowledging methodological heterogeneity that limits cross-study comparisons. We hope that the scoping review findings will encourage researchers in the Middle East to conduct future research considering some of the methodological issues we have highlighted.
The title is misleading. The manuscript does not explore the burden or impact of eating disorders, but rather the prevalence of clinically significant disordered eating. The title should be re-worded to align more specifically with the aims of the manuscript.
Author Response: Thank you for your comment. Another reviewer made a similar comment. In view of these comments, we have amended the term “burden” to “prevalence” and also standardised the use of the term “high-risk disordered eating” as the EAT-26 instrument is a non-diagnostic screening tool.
Minor issues:
Two of the studies reported the prevalence of high-risk DE in a weight loss center and a fitness program, do the authors not believe these estimates may be biased by the recruited population?
Author Response: Thank you for highlighting this observation. We have included two sentences in the ‘Limitations’ section briefly discussing the two studies that recruited participants from a weight loss centre and fitness program (please see Page 11 Lines 298-303).
Table 1 is not formatted consistently. For example, the “Age group/mean age” column is arranged differently across studies. There is no definition provided for “T” under the “Prevalence of High-Risk Disordered Eating” section. It appears the authors meant to include the type of assessment in the “Clinical Assessment” column but that information was not provided. Further, the type of clinical assessment used seems necessary to specify.
Author Response: We thank Reviewer 2 for highlighting our inconsistencies and omission and we have amended Table 1 accordingly. In addition, we have included the clinical tools used to quantify the prevalence of high-risk disordered eating. “T” is defined in the footnote as “Total” and the type of clinical assessment has been added.
Line 59 and 60 mention a higher prevalence is suggested in female adolescents, but the evidence provided indicates a similar range for males and females.
Author Response: We thank Reviewer 2 identifying our error and we have amended the sentence to reflect the data presented.
Research demonstrating that the EAT-26 and EAT-40 have been validated for use in Arabic, Turkish, Hebrew, and Persian languages need to come sooner in the manuscript and be described in more detail. Differences in prevalence may be due to differences in interpretation of the questionnaire items.
Author Response: Reviewer 2 raises an interesting point about the potential differences in interpretation of the questionnaire items in different languages. In view of this insightful comment, we have added a sentence discussing this on Page 10 Lines 239-241 and Page 11 Lines 303-308. We are satisfied with the current location of the text describing questionnaire validation in other languages as it relates to the points made regarding interpretation of questionnaire items in different languages.
It is unclear why “records excluded n= 25 papers” in Figure 1 were excluded. Could more detail be added to the figure? Further, 6 papers were excluded due to not using the EAT-26 or EAT-40; however, those papers could be included if the scales used also have a validated clinical cut-off score.
Author Response: We thank Reviewer 2 for identifying our omission and Figure 1 has been updated. The a priori inclusion criteria was to only include studies that used either the EAT-26 or EAT-40. We hope that our scoping review motivates other researchers to build upon our work with further studies and systematic reviews.
Reviewer 3 Report
Please see the attached document with the comments to the authors.

Author Response
Reviewer 3
Thank you for inviting me to review this article. This article concerns a scoping review on the prevalence of eating disorders among adolescents and young adults in Middle East. To date, no review has been conducted on this topic so the paper covers a gap in the literature. However, in my opinion, the paper needs a major, thorough revision. Please consider addressing the comments for each section considered below.
Author Response: We thank Reviewer 3 for taking the time to review our manuscript. The insightful feedback has helped us to improve our manuscript and we hope that you think the revised version makes a useful addition to the sparse literature on high-risk disordered eating in the Middle East region.
Comments by area:
INTRODUCTION
- In the first paragraph the authors state “However, using multimedia technologies and poor body image perception are considered to be predisposing factors [4]…” I would invite the authors to be more cautious selecting the predisposing factors for eating disorders (EDs) as they are indeed many, equally relevant, biological, psychological, developmental, and sociocultural risk factors. I would reformulate this statement acknowledging the multiple risk factors involved in the etiology of EDs.
Author Response: We agree with Reviewer 3 and we have amended the sentence in view of this comment.
- The paper addresses specifically EDs among adolescents and young adults, however, no rationale for choosing such developmental periods is provided. It would be very positive to include that, for instance, as these are the developmental periods when EDs peak. The same for including both genders.
Author Response: Thank you for your comment. We have included a statement on adolescence and young adulthood as an important risk factor for ED. The rationale for both genders is provided with the results from the ARAB-EAT project that reported a similar prevalence of EDs amongst males and females (Page 2 Lines 59-64) and we have added text and references for the crucial developmental periods and early detection (Page 2 Lines 64-65 and Lines 69-73).
- The authors may include as well the clinical relevance of assessing the prevalence of disordered eating (EDs at subclinical level).
Author Response: Thank you for your input. We have added clinical relevance of detecting ED early and added references (Page 2 Lines 69-73).
- In the second paragraph, please provide a reference supporting the rationale for selecting the EAT-40 or -26 questionnaire for the first stage. As far as I know there are other questionnaires that could assess the presence of EDs or disordered eating. Also please acknowledge that the EAT has many limitations, especially around its validity for assessing disordered eating in males.
Author Response: Thank you for your comment. We have added the rationale for using the EAT-26/40 questionnaire (Page 4 Lines 142-149) and the limitations on Page 11 Lines 294-298 and Lines 304-312).
- Also, provide a reference supporting the “gold standard method” for assessing the risk of and presence of EDs.
Author Response: We have added the reference on Page 2 Line 50.
- The authors conducted a scoping review. I would appreciate more explanation about what a scoping review is, differences from a systematic review of the literature and why such approach was chosen for conducting this study.
Author Response: Thank you for your comment. The primary purpose of a scoping review is to synthesize the available evidence on a specific topic, highlight the gaps, and identify the limitations. A scoping review is preferred when the extant scientific literature on a particular topic has not been comprehensively reviewed and/or is heterogeneous in nature and not suitable for a more precise systematic review. We think that our scoping review synthesizes interesting estimates on the prevalence of high-risk disordered eating in the Middle East whilst acknowledging methodological heterogeneity that limits cross-study comparisons. We hope that the scoping review findings will encourage researchers in the Middle East to conduct future research considering some of the methodological issues we have highlighted.
METHOD
- The biggest issue I find in this section is that there is no protocol pre-registered of this work. Please add this aspect as Limitation of the study.
Author Response: We thank the Reviewer for their comment. Whilst we support the pre-registration and publication of protocols for systematic reviews, we did not feel it was necessary for this scoping review and PROSPERO does not accept scoping reviews. In hindsight, it would have provided full transparency on our scoping review if we had found a suitable outlet for registering and/or publishing our study protocol. We have added this as a limitation and suggestion on Page 11 Lines 313-320.
- Eligibility criteria: if the focus of the study was on adolescents and young adults, why was there any limitation regarding age?
Author Response: The limited number of studies combined with the fact that many studies recruited heterogenous population in terms of age (adolescent, young adults and older adults) and did not disaggregate the prevalence estimates would have narrowed our scope. We have included that as a limitation on Page 11 Lines 299-301.
- It would be helpful for the reader to explicitly state that studies had to use the “gold standard method” for assessing risk of and presence of EDs to be included; that is, use EAT-40 or -26 AND interview.
Author Response: Thank you for your comment. Given the nature of scarcity of studies on ED prevalence. Our scoping review of high risk disordered eating included studies that only used EAT-26 or EAT-40 with or without clinical assessment. We have added clarity on this in the Methods (Page 3 Lines 90-91). Finally, we have highlighted in our discussion that most studies lacked the gold standard of assessing ED prevalence in the Middle East.
- Exclusion criteria: It would be very positive to provide the rationale for including only studies using EAT-40 or -26.
Author Response: Thank you for your comment. We have added text on Page 4 Lines 142-149.
- Information sources and literature search: One of the search terms included in the search strategy was “Feeding and Eating Disorders”, however, I understood that the authors are also evaluating the prevalence of disordered eating, which is at subclinical level of EDs. It would be very informative for the reader if you could specify which terms are included when entering such search term.
Author Response: As advised by our University research librarian, we conducted a broad literature search to minimize the likelihood of missing evidence. We have added text to clarify this in the Methods (Page 3 Lines 110-112).
- How was the screening performed? Did the authors use any software (e.g., Ryaan, Covidence) for that?
Author Response: We had initially planned to use Covidence software for article storage and the screening process. However, due to the low volume of studies to be screened, we were advised by our research librarian to perform the screening manually and to save the Covidence licences for larger systematic reviews with higher volumes of articles to be screened. We have clarified that the screening was manual on Page 3 Line 119 and added text on the use of systematic review management software in the Discussion (Page 12 Lines 313-320).
- How were the studies stored? Using EndNote, Zotero...? Please provide such information to guarantee that the review was handled correctly.
Author Response: Please see the above response. We have added text on the use of systematic review management software in the Discussion (Page 12 Lines 313-320).
- Study section: How were duplicates removed? Specify the software used. What happened in case of disagreements? How were they solved? A third co-author was invited for discussion?
Author Response: We have clarified that the screening was manual on Page 3 Line 119 and please see the response to point 12. We have added text on the use of systematic review management software in the Discussion (Page 12 Lines 313-320). A third reviewer (psychiatrist) was used to resolve conflicts and this has been clarified in the Methods (Page 3 Lines 121-122).
- Data collection Process and Data Items: How was the data exactly extracted? By whom?
Author Response: We have briefly described the data extraction process in the Methods (Page 3 Lines 124-126).
- The EAT: Including more specificity about the validity and reliability of the questionnaire would be very positive. Also, please describe the three subscales included in the EAT.
Author Response: We have added additional information on the validity, reliability, sub-scales and cut-offs used (Page 4 Lines 142-149).
RESULTS
- What are the additional sources for identifying the two other studies?
Author Response: We also searched Internet search engines and hand-searched the reference lists of previous systematic reviews on the topic and of all included studies. We have included this information in the Methods (Page 3 Lines 116-117). The two additional potentially relevant studies were identified through internet search engines and then subsequently excluded during the initial screening phase.
- Age range of participants were between 10 and 51 years, however, the study is only for identifying EDs prevalence among adolescents and young adulthood. Please, this issue should be addressed. For instance, acknowledge which study included participants over young adulthood.
Author Response: We thank Reviewer 3 for their comment about the age range of the participants in the included studies. The age group and/or mean age (when available) are presented in Table 1. We have also mentioned this in the Limitations on Page 11 Lines 299-301.
DISCUSSION
- Some of the information provided about the EAT questionnaire could be used for justifying its use in the Introduction part.
Author Response: We think that there is sufficient information in the Introduction (Page 2 Lines 49-56) and the Methods (Page 4 Lines 142-149) justifying the use of the EAT questionnaire. We are satisfied with the current location of the text in the Discussion describing questionnaire validation in other languages as it relates to the points made regarding interpretation of questionnaire items in different languages.
- I am missing data about the risk of EDs among males in Middle East and what is the beauty standard for them or what would be the main reasons for the onset of disordered eating.
Author Response: Reviewer 3 raises a valid point about the sex differences in the prevalence estimates of high-risk for disordered eating in the Middle East. Where available, we have presented the disaggregated prevalence estimates by sex in Table 1. We agree about the lack of data on the sociocultural drivers of ED development in male adolescents and young adults. We have added a sentence on this in the Discussion (Page 11 Lines 283-286).
- Please, acknowledge the limitations of the EAT, especially for measuring EDs among boys, and other Limitations that may appear after addressing the previous comments.
Author Response: Thank you for your comment. We have added the limitations of the EAT questionnaire on Page 11 Lines 294-298 and Lines 304-312.
Round 2
Reviewer 3 Report
I praise the authors for the work done improving their manuscript. In my opinion, after some minor language editing, the paper is suitable for publication.